Invasion of the tropical earthworm Pontoscolex corethrurus (Rhinodrilidae, Oligochaeta) in temperate grasslands

Ortiz-Gamino Diana 1
Pérez-Rodríguez Paulino 2
Ortiz-Ceballos Angel I. angortiz@uv.mx 1
1 Instituto de Biotecnología y Ecología Aplicada (INBIOTECA), Universidad Veracruzana , Xalapa , Veracruz , México
2 Programa de Estadística, Colegio de Postgraduados-Campus Montecillo , Texcoco , Estado de México , México
Geissen Violette
Electronic publication date: 2016 Oct 12
Publication date: 2016
Volume: 4
Electronic Location ID: e2572
Received 2016 Feb 26; Accepted 2016 Sep 15
Copyright: ©2016 Ortiz-Gamino et al.
Copyright year: 2016
Copyright holder: Ortiz-Gamino et al.
License: This is an open access article distributed under the terms of the Creative Commons Attribution License, which permits unrestricted use, distribution, reproduction and adaptation in any medium and for any purpose provided that it is properly attributed. For attribution, the original author(s), title, publication source (PeerJ) and either DOI or URL of the article must be cited.
License URL: https://creativecommons.org/licenses/by/4.0/

Keywords: Agroecosystems, Soil fauna, Exotic species, Biological invasions

Funding: Consejo Nacional de Ciencia y Tecnología (CONACyT) Mexico 251818 Consejo Nacional de Ciencia y Tecnología (CONACyT) Mexico awarded a PhD scholarship (No. 251818) to Diana Ortiz-Gamino. The funders had no role in study design, data collection and analysis, decision to publish, or preparation of the manuscript.

==============================
The tropical earthworm Pontoscolex corethrurus (Rhinodrilidae, Oligochaeta) presents a broad distribution (e.g., 56 countries from four continents). It is generally assumed that temperature appears to limit the success of tropical exotic species in temperate climates. However, the distribution range of this species could advance towards higher elevations (with lower temperatures) where no tropical species currently occur. The aim of this study was to evaluate the soil and climatic variables that could be closely associated with the distribution of P. corethrurus in four sites along an altitudinal gradient in central Veracruz, Mexico. We predicted that the distribution of P. corethrurus would be more related to climate variables than edaphic parameters. Five sampling points (in the grassland) were established at each of four sites along an altitudinal gradient: Laguna Verde (LV), La Concepción (LC), Naolinco (NA) and Acatlán (AC) at 11–55, 992–1,025, 1,550–1,619 y 1,772–1,800 masl, respectively. The climate ranged from tropical to temperate along the altitudinal gradient. Ten earthworm species (5 Neotropical, 4 Palearctic and 1 Nearctic) were found along the gradient, belonging to three families (Rhinodrilidae, Megascolecide and Lumbricidae). Soil properties showed a significant association (positive for Ngrass, pH, permanent wilting point, organic matter and P; and negative for Total N, K and water-holding capacity) with the abundance of the earthworm community. Also there seems to be a relationship between climate and earthworm distribution along the altitudinal gradient. P. corethrurus was recorded at tropical (LV and LC) and temperate sites (NA) along the altitudinal gradient. Our results reveal that soil fertility determines the abundance of earthworms and site (climate) can act as a barrier to their migration. Further research is needed to determine the genetic structure and lineages of P. corethrurus along altitudinal gradients.

Introduction

Within soil biodiversity, earthworms are key components of the guild of ecosystem engineers (Jones, Lawton & Shachak, 1994). They provide a considerable level of ecosystem services, such as contributing to biogeochemical cycling and crop productivity (Turbé et al., 2010; Orgiazzi et al., 2016). Depending on their ecological classification (epigeic, endogeic or anecic), they can also modify the distribution and abundance of soil biodiversity, mainly by constructing structures and galleries within the soil profile and by producing casts and mucus (Turbé et al., 2010; Orgiazzi et al., 2016).

Most earthworm communities display an aggregated spatial distribution in response to soil environmental heterogeneity at a small scale (Geissen, Peña-Peña & Huerta, 2009; Huerta & Van der Wal, 2012; Jiménez et al., 2012). This is relevant because of the limited capabilities of earthworms for horizontal displacement, between 4 and 10 m per year (Brown et al., 2006; King, Tibble & Symondson, 2008), and may thus have had the opportunity to diversify allopatrically over relatively short distances (Bickford et al., 2005; King, Tibble & Symondson, 2008). Nevertheless, climate has a substantial influence on earthworms (physiology, development or activity) that is reflected in the seasonal dynamics of their life history (Curry, 2004; Turbé et al., 2010; Orgiazzi et al., 2016).

At the global level, apart from studies in North America where non-native earthworms are causing changes in soil biota and plant communities, little recognition has been given to invasions of soil organisms (Gates, 1954; Bohlen et al., 2004; Fahey et al., 2013). Among the 3,700 earthworm species described, approximately 3% (100–120) have been identified as invasive; for example, the ubiquitous Pontoscolex corethrurus (Müller, 1857), several temperate species from genera Amynthas and at least 10 species of Lumbricid (Brown et al., 2006; Beddard, 1912; Hendrix et al., 2008; Dupont et al., 2012). These earthworms have reached a broad distribution in many tropical and temperate agroecosystems and natural ecosystems. However, this has been facilitated by fishing bait, horticulture, waste management industries, road networks and vehicle transport which have contributed to surmount important biogeographic barriers (Eisen, 1900; Beddard, 1912; Gates, 1954; Hendrix et al., 2008).

P. corethrurus “Brush-tail” is native to the Guiana Shield area of the Amazon (Müller, 1857; Brown et al., 2006; Dupont et al., 2012). Due to a high level of genetic diversity in populations (Dupont et al., 2012; Cunha et al., 2014), its adaptive strategies include a high tolerance to soil conditions and climatic variables (precipitation) (Lavelle et al., 1987; González et al., 2006; González et al., 2007). For this reason, P. corethrurus is established throughout the tropical regions of over 56 countries (González et al., 2006). Given the wide distribution range of P. corethrurus, this species can be used as a model organism to investigate and compare the effects of native and introduced species on ecosystem processes. P. corethrurus generally plays a beneficial role in soils; that is, it has the potential to improve plant growth and productivity (Senapati et al., 1999; Van Groenigen et al., 2014; Cunha et al., 2016), and is used as a key indicator in toxicological research (Zavala-Cruz et al., 2013; García-Pérez et al., 2014). There are very few cases of negative or null effects of P. corethrurus (Brown et al., 2006); but soil compaction may be induced under particular situations (Gates, 1954; Chauvel et al., 1999; Barros et al., 2004).

In Mexico, since the early twentieth century, P. corethrurus has been the endogeic earthworm most commonly found in human-altered tropical ecosystems (Eisen, 1900; Lavelle et al., 1987; Brown et al., 2004; Fragoso & Rojas, 2014). However, the edge of the earthworm’s distribution range could advance towards higher elevations where few or no tropical species currently occur (Eisen, 1900; Beddard, 1912; Hendrix et al., 2008). The aim of this study was to evaluate the soil and climatic variables that could be closely associated with the distribution of P. corethrurus. We predicted that the distribution of P. corethrurus would be more related to climatic variables than edaphic parameters. We tested this hypothesis through a study of the earthworm community along an altitudinal gradient in central Veracruz, Mexico. In addition, we compared the occurrence of four possible situations in the altitudinal gradient (Marichal et al., 2010): (a) presence of P. corethrurus only, (b) coexistence of P. corethrurus and other species (native and invasive), (c) absence of P. corethrurus but presence of other species (native and exotic), and (d) absence of earthworms.

Methods

Study area

An altitudinal transect, ranging from 11 to 1,800 masl, was established in the central region of the State of Veracruz, Mexico. Five sampling points were established at each of four sites along this altitudinal gradient (Fig. 1): Laguna Verde (LV), La Concepción (LC), Naolinco (NA) and Acatlán (AC) at 11–55, 992–1,025, 1,550–1,619 and 1,772–1,800 masl, respectively.

Figure 1 Sampling sites of earthworms along an altitudinal gradient in central Veracruz, Mexico. For each site, the geographical coordinates (14N zone, Datum WGS84) are presented.

LV, Laguna Verde (11–55 masl); LC, La Concepción (992–1,025 masl); NA, Naolinco (1,550–1,619 masl); AC, Acatlán (1,772–1,800 masl). Digital elevation model created using the geographical data provided by Instituto Nacional de Estadística y Geografía, Mexico.

Climate information from the Mexican National Water Commission weather stations (http://www.conagua.gob.mx) was compiled for each site along the altitudinal transect. The monthly and annual 30-year averages were obtained for the following climate variables: average temperature (AT), average maximum temperature (AMT), average minimum temperature (AmT), total annual precipitation (TAP) and total evaporation (TE). With these data, climate types along the altitudinal gradient were determined using the Clima2 software (http://www.pablo-leautaud.com/home/proyectos/python/clima) and classified into one climate type according to the Köppen-Geiger system (Kottek et al., 2006; Peel, Finlayson & McMahon, 2007).

Earthworm sampling

For determining the distribution of earthworms, we choose the grasslands because they are a suitable habitat that fosters the growth of earthworms, and can be found along the altitudinal gradient. These grasslands are used as pasture for both an extensive (LV and LC) and semi-intensive cattle farming (NA and AC) under a rotational use of pasture (30–45 day grazing-rest cycle) and without the application of mineral fertilization (Lavelle, Maury & Serrano, 1981; Brown et al., 2004). The grasses species that grow in these grassland are: (a) native: sour paspalum Paspalum conjugatum P.J. Bergius (80% LV, 60% LC, 40% NA, and 40% AC), (b) introduced: bermudagrass or stargrass Cynodon nlemfuensis Vanderyst (20% LV, 40% LC, and 40% NA) and kikuyu Pennisetum clandestinum Hochst. Ex Chiov. (20% NA, and 60% AC). In the extensive production system (Dual-purpose cattle systems: milk and meat), cattle (Bos indicus × Bos Taurus cows: Zebu × Swiss or Holstein) was fed only on forage produced in grassland, ocassionaly suplemented with mineral salts. By contrast, under the semi-intensive system (milk), besides feeding on grassland forage, milk cows (Holstein) are given a dietary supplement made of carbohydrates (corn and barley stubble), protein (cane molasses, urea, dehydrated alfalfa and others) and mineral salts.

The quantitative sampling of earthworms was conducted along the altitudinal gradient (International Organization for Standarization, 2011). One monolith (25 × 25 × 30 cm deep) according to Anderson & Ingram (1993) was sampled at each of the five sampling points (in the grassland) established at least 200 m apart (Marichal et al., 2010), located at each of the four sites on the altitudinal transect, for a total of 20 monoliths along the altitudinal transect. Each monolith was separated into four strata: above-ground plant biomass, 0–10, 10–20 and 20–30 cm. Earthworms were then manually removed from each layer and preserved in 70% ethanol. In the laboratory, all specimens were fixed in 4% formaldehyde and then identified (to species or morphospecies), quantified and weighed. The sampling was conducted at the end of the rainy season (August–October 2011). Abundance and biomass data of the earthworms were converted into densities per square metre (ind. m−2 and g m−2, respectively) for each site (International Organization for Standarization, 2011).

Soil and foliage sampling

Prior to removing the earthworms, the biomass of grass was harvested from each monolith. In the laboratory, this plant material was dried (60° C for 72 h) and weighed, and its total nitrogen content (Ngrass) was determined using the Kjeldahl methods described in the Mexican Official Standard NOM-02-RECNAT-2000 (SEMARNAT, 2002).

Following the removal of earthworms, a 1-kg soil sample was taken from each stratum of each monolith. Soil samples were air-dried to constant weight and sieved (5 mm) to determine the texture (clay, silt and sand), water-holding capacity, permanent wilting point, pH, organic matter, Total C, Total N, P and K, using the methods described in the Mexican Official Standard NOM-02-RECNAT-2000 (SEMARNAT, 2002).

Statistical analysis

A one-way ANOVA was used to test for significant differences (P < 0.05) in soil properties between sites, using the Statistica software, ver. 7 (StatSoft, Tulsa, OK, USA).

We used a Linear Model (LM) to study earthworm abundance (ind. m−2) using soil properties and climatic elements. The effect of the climatic elements was included in the model through the variable “site” because it was possible to clearly distinguish the sites in a scatter plot (figure not shown) of the scores for the first two principal components obtained from a PCA of the climatic variables. The dependent variable (earthworm abundance) was transformed using natural logarithms because the empirical distribution of earthworms was highly asymmetric. All the analyses were performed using R software (R Core Development Team, 2015). We also fitted Linear Mixed Models to take into account the sampling design consisting in clustered samples and the response variable measured at two different scales, i.e., soil properties at the sample scale and climatic conditions at the site scale. The results showed that the variance component associated with the site effect was misleading. Apart from the linear models and before log-transforming the data, we also fitted generalized linear models with different families (e.g., Poisson, Negative Binomial); however, some convergence issues arose when fitting the models. Consequently, only the results for the fitted Linear Model are reported here.

Results

Site climate

The climate along the altitudinal gradient, from the lowest to the highest elevation sites, according to Kottek et al. (2006) and Peel, Finlayson & McMahon (2007) ranged from warm to humid tropical (Aw) to temperate (Cfb) (Table 1). There was a difference of approximately 11°C in the average, minimum and maximum temperatures between the lowest (LV, 11–55 masl) and the highest elevation sites (AC, 1,772–1,800 masl) along the gradient. Rainfall was higher in site LC (1676.8 mm) than in site LV (1143.0 mm), whereas sites NA and AC (1,550–1,619 and 1,772–1,800 masl) had intermediate values of 1,461 mm (Table 1).

Table 1 Climate variables at the four sampling sites along an altitudinal gradient in central Veracruz, Mexico.

Site	Altitude	Temperature (° C)	TAP	TE	Month most	
	(masl)	AT	AMT	AmT	(mm)	Warmer	Cooler	
LV	11–55	26.0 ± 2.5	30.0 ± 2.5	21.0 ± 2.0	1143.0	1618.1	June	January	
LC	992–1,025	20.0 ± 2.0	26.0 ± 2.2	14.0 ± 1.8	1676.8	1322.0	May	January	
NA	1,550–1,619	17.0 ± 2.0	22.2 ± 2.0	12.0 ± 1.9	1462.0	1554.8	May	January	
AC	1,772–1,800	15.0 ± 1.8	20.0 ± 1.9	10.0 ± 1.8	1461.0	1190.8	May	January	
Notes.

LV Laguna Verde

LC La Concepción

NA Naolinco

AC Acatlán

AT Average temperature

AMT Average maximum temperature

AmT Average minimum temperature

TAP total annual precipitation

TE total evapotranspiration

Soil properties and foliage

The physical and chemical variables of soil and nutritional quality of pasture along the altitudinal gradient displayed significant variations between tropical (LV and LC) and temperate (NA and AC) sites (Table 2). According to and the official Mexican standard soil fertility (SEMARNAT, 2002), soils from tropical sites had a heavy texture (clay loam; Regosols, Phaeozems and Vertisols; (Krasilnikov et al., 2013); were mildly acidic; and displayed intermediate values for water-holding capacity and permanent wilting point; were very rich in organic matter, total N and P; and were extremely poor in K. In contrast, soils from temperate sites had a light texture (loam; Andosols); greater water-holding capacity and permanent wilting point; were slightly acidic; and were very rich in organic matter, total N and P; and were extremely poor in K. Quality grass (Ngrass) in the temperate sites (NA and AC) was higher compared to tropical sites (LC an LC).

Table 2 Soil properties and grass at the four sites along an elevation gradient in central Veracruz, Mexico.

Parameter	LV	LC	NA	AC	
Clay, %	27.6 b	25.6 b	12.0 a	13.6 a	
Silt, %	21.8 a	34.4 b	28.8 ab	35.8 b	
Sand,%	50.6 ab	40.0 a	59.2 b	50.6 ab	
pH, (H2O)	6.6 b	6.6 b	5.5 a	5.6 a	
Water-holding capacity, %	32.9 a	36.6 a	83.5 b	70.5 b	
Permanent wilting point, %	20.0 a	20.2 a	53.0 b	40.4 b	
Organic matter, %	8.0 a	8.4 a	43.7 b	34.2 b	
Total N, %	0.31 a	0.31 a	1.13 c	0.83 b	
C/N	15 a	16a	22 b	24 b	
P, mg kg−1	13.2 a	54.7 c	49.8 b	14.2 a	
K, cmolc kg−1	1.1 a	1.7 a	2.8 b	1.4 a	
Ngrass, %	1.1 a	0.71 a	1.46 b	1.45 b	
Notes.

Within each row different letters indicate significant differences at P < 0.05 (Tukey’s HSD test).

LV Laguna Verde (11–55 masl)

LC La Concepción (992–1,025 masl)

NA Naolinco (1,550–1,619 masl)

AC Acatlán (1,772–1,800 masl)

Earthworm communities

Ten earthworm species (Annelida: Oligochaeta: Crassiclitellata) were found in the whole sampling (Table 3). Seven of these are well-known, ubiquitous species, some of which are considered invasive, belonging to three different families (Rhinodrilidae, Megascolecidae and Lumbricidae). The remainder of the earthworms were native morphospecies (differentiated from others only by morphological features). The highest diversity was found at site AC, with five species. The total abundance of the earthworm community ranged from 0 to 864 ind. m−2 (Fig. 2), with an average of 332 ind. m−2.

Table 3 Earthworm species recorded in four sampling sites along an altitudinal gradient in central Veracruz, Mexico.

Species	Family	Origin	Parthenogenetic	Ecological category	Sites and average density (ind./m2)	
					LV	LC	NA	AC	
Pontoscolex corethrurus	Rhinodrilidae	South America	+	Endogeic	141	330	144	0	
Onychochaeta windlei	Rhinodrilidae	South America	+	Endogeic	0	320	0	0	
Morph 1	Morphospecies	Mexico	Uncertain	Endogeic	264	0	0	0	
Morph 2	Morphospecies	Mexico	Uncertain	Endogeic	155	0	0	0	
Morph 3	Morphospecies	Mexico	Uncertain	Endogeic	160	0	0	0	
Amynthas gracilis	Megascolecidae	Asia	+	Epi-endogeic	0	240	53	80	
Octolasion tyrtaeum	Lumbricidae	Europe	−	Endogeic	0	0	120	67	
Aporrectodea trapezoides	Lumbricidae	Europe	+	Endogeic	0	0	0	128	
Lumbricus rubellus	Lumbricidae	Europe	−	Epi-endogeic	0	0	0	72	
Bimastos parvus	Lumbricidae	North America	+	Epigeic	0	0	0	21	
Notes.

LV Laguna Verde (11–55 masl)

LC La Concepción (992–1,025 masl)

NA Naolinco (1,550–1,619 masl)

AC Acatlán (1,772–1,800 masl)

Figure 2 Abundance (A) and proportion (B) of Pontoscolex corethrurus and total earthworm community abundance (C) along an altitudinal gradient.

LV, Laguna Verde (11–55 masl); LC, La Concepción (992–1,025 masl); NA, Naolinco (1,550–1,619 masl); AC, Acatlán (1,772–1,800 masl).

The LM analysis showed that the total abundance of the earthworm was significantly influenced by water-holding capacity (P = 0.026), permanent wilting point (P = 0.019), pH (P = 0.045), organic matter (P = 0.029), Total N (P = 0.015), P (P = 0.031), K (P = 0.016) and Ngrass (P = 0.009), while the climatic factors (sites) had no such effect (F = 5.57; P = 0.0652). That is, positive coefficients were associated with an increase in the number of earthworms, and negative coefficients were associated with a decrease in the number of earthworms (Table 4).

Table 4 Estimated regression coefficients in the linear model that predict total earthworm abundance along an altitudinal gradient in central Veracruz, Mexico.

Factors	Estimate coefficients	Std. Error	t	P	
Intercept	44.410	1882.0	0.024	0.982	
Site AC*	−0.394	0.792	−0.497	0.645	
Site LC*	0.626	0.554	1.130	0.322	
Site NA*	−0.324	1.095	−0.296	0.782	
Clay, %	−0.133	0.067	−1.98	0.118	
Sand, %	0.001	0.034	0.06	0.957	
Water-holding capacity, %	−0.208	0.061	−3.43	0.026	
Permanent wilting point, %	0.484	0.128	3.77	0.019	
pH, (H2O)	0.920	0.322	2.86	0.045	
Organic matter, %	0.156	0.044	3.32	0.029	
Total N, %	−18.090	4.420	−4.01	0.015	
P, mg kg−1	0.018	0.006	3.26	0.031	
K, cmolc kg−1	−1.053	0.263	−4.01	0.016	
Ngrass, %	3.559	0.770	4.62	0.009	
Notes.

Sites:

LV Laguna Verde (11–55 masl)

LC La Concepción (992–1,025 masl)

NA Naolinco (1,550–1,619 masl)

AC Acatlán (1,772–1,800 masl)

* The “site” was included in the linear model using Dummy variables with the reference cell method. The reference site was LV. We performed an analysis of variance for this linear model; the P-value associated to the site effect was 0.0652.

Pontoscolex corethrurus

Populations of P. corethrurus were found in 10 of the 20 samples from the gradient (Fig. 2A): LV (1/5), LC (5/5) and NA (4/5), but the species was absent in all samples of site AC (situated at 1,772–1,800 masl).

On average, the abundance of P. corethrurus accounted for 73% of the total earthworm density throughout the samples where the species was present. This percentage varied between sites LV, LC and NA at 92, 79 and 47%, respectively (Fig. 2B). In the sites where the species occurred, its average density was 273.5 ind. m−2, ranging from 16 to 704 ind. m−2 (Fig. 2C).

Pontoscolex corethrurus coexisted with exotic (two and four of the five samples in LC and NA, respectively) and native (1/5 in LV) species (Table 5). In contrast, P. corethrurus was found alone in three of the five monoliths of site LC, while only native species were found alone in site LV.

Table 5 Earthworm community composition in each of the five monoliths at the four sites along an altitudinal gradient in central Veracruz, Mexico.

no earthworms, Pontoscolex corethrurus only, coexistence (exotic and native), others species (but no P. corethrurus).

Sites	No earthworm	P. corethrurus only	Coexistence	Others species only	Total	
			Exotic	Native	Exotic	Native		
LV	0	0	0	1	0	4	5	
LC	0	3	2	0	0	0	5	
NA	1	0	4	0	0	0	5	
AC	0	0	0	0	5	0	5	
Total	1	3	6	1	5	4	20	
Notes.

LV Laguna Verde (11–55 masl)

LC La Concepción (992–1,025 masl)

NA Naolinco (1,550–1,619 masl)

AC Acatlán (1,772–1,800 masl)

Discussion

Earthworm communities are determined by hierarchical organized factors: temperature operates at the highest hierarchical level, followed by soil nutrient and seasonality factors (Gerard, 1967; Fragoso & Lavelle, 1992; Briones et al., 2009; Eisenhauer et al., 2014). Compared to other types of terrestrial ecosystems, grasslands (which are the best carbon storage systems) are structurally simple and appear to be relatively homogeneous in terms of richness and functional complexity, particularly belowground (Stockdill, 1966; Stanton, 1988; Brown et al., 2004). Here, we found that along an altitudinal gradient, site (climate) can act as a barrier to distribution of peregrine earthworms and their abundance was determined significantly by soil fertility and grass quality.

Earthworm community

Along the altitudinal gradient studied here, 10 species (seven exotic and three native morphospecies) were recorded in the grassland. The exotic species are among the 51 exotic species recorded in Mexico, and the three morphospecies are among the 40 native species that are already known but still undescribed (Fragoso & Rojas, 2014). Of the ten species that we found, five are Neotropical (P. corethrurus, Onychochaeta windelei and three morphspecies), three are Western Palearctic (Lumbricus rubellus, Aporrectodea trapezoides and Octolasion tyrtaeum), one is Nearctic (Bimastos parvus) and one more is Eastern Palearctic (Amynthas gracilis). The earthworm diversity was similar (five species) between the tropical (LV and LC) and temperate sites (NA and AC), similar to the diversity (4–14 species) that has been observed in tropical and temperate forests (Fragoso & Lavelle, 1992).

The current state of knowledge allows little generalization about the distribution patterns of invasive earthworms, as is the case of P. corethrurus (Hendrix et al., 2008). However, Beddard (1912) suggested that: (a) temperate species tend to invade temperate regions and montane areas of the tropical regions, and (b) tropical earthworms only tend to invade tropical regions; that is, low temperatures limit their colonization of temperate areas. Our results show a trend for the influence of climate (site) on the distribution of earthworm species throughout the altitudinal gradient: the Palearctic and Nearctic species where only found in the temperate sites (NA and AC) and the Neotropical species only in the tropical sites (LV and LC). However, P. corethrurus and A. gracilis was found in one temperate (NA), in one tropical (LC) site, respectively. Some characteristics of native species (morph 1 and 3) and their soil habitats (e.g., high content of expending clay) might be resistant to introduction of P. corethrurus in the site LV, that is, because their presence around this site was registered 35 years ago (Lavelle, Maury & Serrano, 1981). Also just as recorded by Juárez-Ramón & Fragoso (2014) and this study (site AC) P. corethrurus does not coexist with L. rubellus and A. trapezoides. In contrast, P. corethrurus coexists with Palearctic (A. gracilis and O. tyrtaeum) species as observed by Huerta, Gaspar-Genico & Jarquin-Sanchez (2014) and Juárez-Ramón & Fragoso (2014), respectively. Furthermore, P. corethrurus coexist with Neotropical (morph 2 and O. windelei) species.

Grasslands have a positive effect on earthworm diversity and abundance, which is in line with known habitat preferences (Stockdill, 1966; Stanton, 1988; Brown et al., 2004; Rutgers et al., 2016). In this study the diversity and density of earthworms along the altitudinal gradient (ranging from 1–10 and 80–864 ind. m−2, respectively) fall within the range reported (1–35 and 6-850 ind. m−2, respectively) for grasslands, croplands and forests of different tropical and temperate regions of Mexico (Fragoso & Lavelle, 1992; Ordaz, Barois & Aguilar, 1996; Brown et al., 2004; Huerta et al., 2007; Geissen, Peña-Peña & Huerta, 2009; Huerta & Van der Wal, 2012; Uribe et al., 2012; Lavelle et al., 1987; Fragoso, Coria-Martínez & Camarena, 2009; Juárez-Ramón & Fragoso, 2014), Puerto Rico (González et al., 2007), Colombia (Feijoo et al., 2010; Marichal et al., 2010; Gutiérrez-Sarmiento & Cardona, 2014), Brasil (Marichal et al., 2010), USA (Fahey et al., 2013; Eisenhauer et al., 2014), Canada (Eisenhauer et al., 2007) and Europe (Rutgers et al., 2016).

In grasslands, the net primary productivity and the secondary productivity take place in the soil: the former as rhizosphere (root and exudates), and the latter as soil biota (Stockdill, 1966; Stanton, 1988; Trujillo, Fisher & Lal, 2005). The higher organic matter, N and P content in soils across the altitudinal gradient derives from (Beetle, 1974; Trevaskis, Fulkerson & Nandra, 2004; Wright, Hons & Rouquette Jr, 2004; Trujillo, Fisher & Lal, 2005; Mislevy & Martin, 2006; Jones, Orton & Dalal, 2016): (a) the higher productivity and nutrient quality of introduced grasslands (C. nlemfuensis and P. clandestinum), (b) carbohydrate and protein supplementation (NA and AC), and (c) low cattle density (one animal per hectare). This allows to incorporate a higher amount of leaf and root litter, dung and urine into soil; that is, well-managed grasslands have the potential to sequestrate large amounts of carbon and nutrients (Lal, 2004; Wright, Hons & Rouquette Jr, 2004; Jones, Orton & Dalal, 2016). Also, our results, showed that the total abundance of earthworms, irrespective of the species is explained by foliage quality (nitrogen) in the pasture, and soil fertility; that is, we found that Ngrass, organic matter, pH, P and permanent wilting point increase abundance, while total N, K and water-holding capacity apparently reduce it. This has also been documented in other field (Zou & González, 1997; Brown et al., 2004; González et al., 2007; Geissen, Peña-Peña & Huerta, 2009; Huerta & Van der Wal, 2012; Huerta & Van der Wal, 2012; Jiménez et al., 2012; Uribe et al., 2012) and laboratory studies (Patrón et al., 1999; Ganihar, 2003; Marichal et al., 2012). Soil organic matter is an important ‘building block’ for soil structure, to absorb water, to retain nutrients and for aeration (Turbé et al., 2010). Also, earthworms have higher abundance in soils with pH from 5 to 7 (LV, LC, NA and AC) and P is maximised when it is in this pH range (Lavelle et al., 1987; González et al., 2007; Marichal et al., 2010). Also, soil with high silt contents (LV, LC, NA and AC) is a favourable habitat for earthworms (Huerta et al., 2007; Geissen, Peña-Peña & Huerta, 2009; Marichal et al., 2010; Huerta & Van der Wal, 2012). However, abundance of earthworms is directly affected by the reduction of soil organic matter content and indirectly by the reduction in plant diversity and productivity (Brown et al., 2004; Brown et al., 2006; Marichal et al., 2010).

P. corethrurus

Our results showed that the most common species, with the exception of AC, along an altitudinal gradient was the endogeic P. corethrurus. Several field studies in different tropical regions of Mexico (Lavelle et al., 1987; Ordaz, Barois & Aguilar, 1996; Brown et al., 2004; Huerta et al., 2007; Fragoso, Coria-Martínez & Camarena, 2009; Geissen, Peña-Peña & Huerta, 2009; Huerta & Van der Wal, 2012; Uribe et al., 2012; Juárez-Ramón & Fragoso, 2014) and other tropical countries (Zou, 1993; Hallaire et al., 2000; Barros et al., 2004; Zou & González, 1997; González et al., 2007; Feijoo et al., 2010; Fonte & Six, 2010; Marichal et al., 2010) have documented that P. corethrurus populations (from 0 to 804 ind. m−2) are only found in environments with an average annual temperature of 24.1 ± 3.9°C (range: 16–33°C), similar to sites LV and LC that show average earthworm densities of 140 and 329 ind. m−2, respectively. However, our findings reveal that the site LV is characterized by the absence of P. corethrurus in soils with a higher clay content (4:5 monoliths) and its presence associated with higher sand content (1:5 monoliths); similar findings were observed by Lavelle, Maury & Serrano (1981) and Fragoso, Ángeles & Cruz (2006) in LV: Palma Sola (clayey) and La Mancha (sandy), respectively. Contrary to the remarks by Buch et al. (2011), our results suggests that soils with a heavy texture (predominance of expandable clays) restrains the survival, growth and movement (for instance, by its brush-tail) of P. corethrurus, since soil is characterized by: (a) humidity levels delimited between hydric stress (high temperature) and excess water (flooding), (b) adhesivity when wet, and (c) hardness when dry. Furthermore, our results suggest that the native morphospecies recorded prevent the establishment of P. corethrurus, similar to the findings reported by Lavelle, Maury & Serrano (1981) and Fragoso (2011) in clayey soils in LV (Palma Sola), where only native species were found: Ramiellona sp. Nov24, Lavellodrilus parvus, Larsonidrilus microscolecinus, Diplotrema sp. Nov9 y Diplocardia sp. Nov4. Marichal et al. (2010) suggest that P. corethrurus and the native species respond to different sets of conditions with variations that are independent between them. Thus, the biological resistance of native communities has been postulated as a key element to understand the presence of P. corethrurus in some habitats bot not in others (Ortiz-Ceballos et al., 2005; Marichal et al., 2012).

In addition, Eisen (1900) states that P. corethrurus “possesses a great vertical range. I do not think, however, that it occurs in localities subject to frost” We observed populations of P. corethrurus (with an average of 133 ind. m−2) at 1550–1619 masl (site NA), where the average annual temperature is 17°C. In San Jerónimo Tecoatl (Oaxaca, Mexico; Juárez-Ramón & Fragoso, 2014), Drakensberg (KwaZulu-Natal, South Africa; Plisko, 2001; Janion-Scheepers et al., 2016), Antsirabe region of Madagascar (Chapuis-Lardy et al., 2010; Villenave et al., 2010), Curitiba (Paraná State, Brazil; Buch et al., 2011), Zipacón (Cundinamarca, Colombia; Gutiérrez-Sarmiento & Cardona, 2014), Chatham (New Jersey, USA; Nearctic region; Gates, 1954) and São Miguel island (the Azores Archipelago; Palearctic region; Cunha et al., 2014), P. corethrurus has become established under similar temperatures (Kottek et al., 2006; Peel, Finlayson & McMahon, 2007; Orgiazzi et al., 2016). This suggests that the growth and reproduction of P. corethrurus may no longer be limited by temperature, as indicated by Lavelle et al. (1987) with ranges of 20–30°C and 23–27 °C, respectively (Zund, Pillai-McGarry & McGarry, 1997; Barois et al., 1999; Patrón et al., 1999; Ganihar, 2003; García & Fragoso, 2003; Topoliantz & Ponge, 2005; Lafont et al., 2007; Chapuis-Lardy et al., 2010; Hernández-Castellanos et al., 2010; Zhang et al., 2008; Villenave et al., 2010; Buch et al., 2011; Chaudhuri & Bhattacharjee, 2011; Marichal et al., 2012; Duarte et al., 2014; Kok et al., 2014; Nath & Chaudhuri, 2014). In these studies, authors have found much faster or slower life cycles, mainly depending on the temperature of incubation and where they were collected (Buch et al., 2011).

Rapid adaptations or mutations in known invasive species should be considered as likely mechanisms that could facilitate their spread into new habitats (Hendrix et al., 2008). Genetic studies have shown a high level of genetic diversity in populations of P. corethrurus and they are probably differentiated into cryptic species (Dupont et al., 2012; Cunha et al., 2014). Our findings suggest that P. corethrurus inhabiting temperate grasslands is a lineage different to sites LV and LC. The correct molecular identification of P. corethrurus is needed to comprehend their history of colonization and as a baseline for biology, ecology and ecotoxicology research on this species (King, Tibble & Symondson, 2008; Dupont et al., 2012; Cunha et al., 2014).

Conclusions

Our results showed that soil quality significantly determined the abundance of the earthworm community along an altitudinal gradient. In addition, climate was shown to be a barrier to distribution of peregrine earthworms as suggested by Beddard (1912). P. corethrurus inhabiting tropical and temperate grasslands probably have 2–3 different lineages or ecotypes. Further studies will be needed to elucidate the genetic diversity of P. corethrurus.

Supplemental Information

Data S1 Raw Data Invasion P. corethrurus

Click here for additional data file.

Data S2 Data (PCA and Linear Mixed Models)

Click here for additional data file.

We thank the farmers in Laguna Verde, La Concepción, Naolinco and Acatlán for allowing access to their properties. Special thank to Carlos Fragoso for support in the identification of earthworms. We are grateful to Rogelio Lara González for technical assistance. We thank Martha Novo and Rosa Fernández for help throughout the study. We also thank Diana Pérez-Staples and two anonymous reviewers for valuable comments and careful revision of the manuscript.

Additional Information and Declarations

Competing Interests

Author Contributions

Data Availability

The authors declare there are no competing interests.

Diana Ortiz-Gamino and Angel I. Ortiz-Ceballos conceived and designed the experiments, performed the experiments, analyzed the data, contributed reagents/materials/analysis tools, wrote the paper, prepared figures and/or tables, reviewed drafts of the paper.

Paulino Pérez-Rodríguez analyzed the data, wrote the paper, prepared figures and/or tables, reviewed drafts of the paper.

The following information was supplied regarding data availability:

The raw data has been supplied as Supplemental Files.

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
