# Peer review of "Invasion of the tropical earthworm Pontoscolex corethrurus (Rhinodrilidae, Oligochaeta) in temperate grasslands"

_PeerJ, doi:10.7717/peerj.2572_

## Round 0.1 · original submission · Major Revisions

Dear authors,

The manuscript presents interesting results, the experimental setup is ok, however, it requires major revision before being published. Please revise the manuscript carefully following the comments of the reviewers and then send it back.

Reviewer 1 ·

Basic reporting

I recommend to revise the English, some sentences are a little strange.
Even though there is a wide literature revision, I encourage the authors to use discuss more the information, and on introduction, to abord the fact that P. corethrurus is a cryptic species, and then it is possible that there is one morphospecies, but with 2-3 genetic lineages. Also it is important to quote works done in Mexico with P. corethrurus, ie: Uribe et al 2012, Huerta et al. 2012, Geissen et al. 2009, Huerta et al. 2007, studies done at field work which show the wide distribution of P. corethrurus to different land use systems, to different altituds(some of them higher than the highest place shown in this study), and these works show also the correlation of this species with soil properties.
I suggest to reformulate the hypothesis, into a simpler one, trying to describe the abundance and diversity of earthworms in communities where P. corethrurus is abundant, along a gradient of altitude.

Experimental design

It is an interesting study, and the design is good if we want to observe which is the distribution of earthworms a long an altitudinal gradient, but for testing if the altitude has an effect over P. corethrurus distribution, I think more transects were required, and also it was required to have the certitude of the type of soil (texture, organic matter content), vegetation (that was ok, all the points were collected in a grassland), and then to have only as factor the altitude. Then if the hypothesis is changed, the experimental design is very ok.
On methods, can you tell more about the history of each site?, for instance the place named Ingenio, was it a sugar cane production area?, and now is it abandoned?, or still it functions as –Ingenio-, this information will allow you to understand more your results, and then will help you to do a better discussion

Validity of the findings

The results, are interesting well shown on tables, but the information is missing on the text, I encourage the authors to exploit more their information, and tell more of it, on the text, also I suggest, may a –path analysis- will be useful for finding the causality on the correlations that they show.
I think you have to condense the discussion, and to focus the discussion on the fact that P. corethrurus has a wide range of distribution, and then you may explain why, what are the possible possibilities and reasons, also you may address the fact, that maybe we have one morphoespecies named P. corethrurus, but that genetically can be 3 species, I encourage you to redo the discussion, in a more concentrate way, its seem that the discussion is divided in two sections, one with generalizations, and one with specifications, I recommend to generalize less, and to concentrate more, focalizing more.
It is true that due to the global change, some worms can be found in some areas, where they have not been found before, but for testing the effect of global change, you need a lot of information, you need to quoted which were the communities of earthworms in those sites at least 50 years ago, in the same areas, with the same vegetation, same soil type, and then to compare with the communities of earthworms found by you, now (by the way, I think you didn’t inform when you had made the collect of earthworms, please add it, thank you).

Additional comments

It is an interesting study, but it requires to add more references of studies done in Mexico with P. corethrurus, ie. Tabasco. It is important to reformulate the hypothesis, to exploit more the results and to condense and concentrate the discussion.

Annotated reviews are not available for download in order to protect the identity of reviewers who chose to remain anonymous.

Reviewer 2 ·

Basic reporting

In general the paper is interesting but some ideas and data need to be specify. It can be publish but with some clarification, more specific data and deeper discussion on the Nitrogen variable and the earthworm community in LV. Figure is missing
For me the work is more a study on earthworm community along an altitudinal gradient than a specific work on Pontosclex corethrurus invasion. So I recommend to change the title. To talk more on P. corethrurus invasion you require more climatic data. The soil temperature in each site also can be a variable that could influence the presence of Pontoscolex corethrurus, do you have some data of it. I believe that in the Azores P. corethrurus is there because the soil is warmer because of the volcanic activity.
You don´t give any description of the grasses. The grasses species from each site are very different? They have the same kind of roots? How old are these grasslands. This can be a factor that influence the earthworm community. In which type of soil were the sites; in LV and IC was it a vertisol and NA an Ac was it an Andosol? General soil maps can give you an idea.
Another point that need more discussion is the Nitrogen: In your Estimated regression coefficients the N total and N grass are the highest by far from the other , thus could be because of fertilizer management or what ?
You say that the earthworm abundance was more influenced by the soil attributes than by the climate. Although the neartic species are only in the temperate sites. Something that you have not discussed and you should is: why in LV there is only in one monolith P. corethrurus and it is the site where the native species are, and they do not coexist with P.corethrurus. What this site has that P. corethrurus is almost not present although the climatic conditions are the best for it? the properties of the soil, the story of the management??. The grass are natural or introduced? The soil texture is sandier. I know that P. corethrurus likes more clayey soil than sandy ones.
The paper need to be revised by a native English speaker
Some points are commented below and in the the manuscript
Kind regards

In the result you put ranges and in the tables you don’t put the specific data from each site. In your tables you should put them as I recommend for table 3 in the text. You should provide a table with the soil properties.
I would not put decimals in the majority of the percentage, only when it is less than 1 %.
Put italics in all the species name
Table 1 is not important. I would prefer that you signal the points in the map. If it is visible as you mention they were separate each at least 200m. I think a table with the data of the characteristics of the soil from each site should be provides as you only give ranges.

Table 3 the proportion of the EW species abundance and the total per site should be mentioned
Table4 the headings are not clear, the first column represent the variables or factors? The * means what? LV is not present you have to show its coefficient or intercept. So the second column is the estimate coefficient. Finally you have to mention the magnitudes and confidence intervals of the fixed effects; the magnitude of the among group variation for each random effect. Or report among group variation as random effect variances or proportions of variance. In the legend mention that these coefficient estimates come from LM o LMM analysis.

Experimental design

The xeperimental design is fine although the statiscal analysis need some clarification.
A climatica variable that should have been measured was the soil temperature.

Validity of the findings

The data give intesresting results and more points to be discussed see general comments

Additional comments

In general the paper is interesting but some ideas and data need to be specify. It can be publish but with some clarification, more specific data and deeper discussion on the Nitrogen variable and the earthworm community in LV. Figure is missing
For me the work is more a study on earthworm community along an altitudinal gradient than a specific work on Pontosclex corethrurus invasion. So I recommend to change the title. To talk more on P. corethrurus invasion you require more climatic data. The soil temperature in each site also can be a variable that could influence the presence of Pontoscolex corethrurus, do you have some data of it. I believe that in the Azores P. corethrurus is there because the soil is warmer because of the volcanic activity.
You don´t give any description of the grasses. The grasses species from each site are very different? They have the same kind of roots? How old are these grasslands. This can be a factor that influence the earthworm community. In which type of soil were the sites; in LV and IC was it a vertisol and NA an Ac was it an Andosol? General soil maps can give you an idea.
Another point that need more discussion is the Nitrogen: In your Estimated regression coefficients the N total and N grass are the highest by far from the other , thus could be because of fertilizer management or what ?
You say that the earthworm abundance was more influenced by the soil attributes than by the climate. Although the neartic species are only in the temperate sites. Something that you have not discussed and you should is: why in LV there is only in one monolith P. corethrurus and it is the site where the native species are, and they do not coexist with P.corethrurus. What this site has that P. corethrurus is almost not present although the climatic conditions are the best for it? the properties of the soil, the story of the management??. The grass are natural or introduced? The soil texture is sandier. I know that P. corethrurus likes more clayey soil than sandy ones.
The paper need to be revised by a native English speaker
Some points are commented below and in the the manuscript
Kind regards

In the result you put ranges and in the tables you don’t put the specific data from each site. In your tables you should put them as I recommend for table 3 in the text. You should provide a table with the soil properties.
I would not put decimals in the majority of the percentage, only when it is less than 1 %.
Put italics in all the species name
Table 1 is not important. I would prefer that you signal the points in the map. If it is visible as you mention they were separate each at least 200m. I think a table with the data of the characteristics of the soil from each site should be provides as you only give ranges.

Table 3 the proportion of the EW species abundance and the total per site should be mentioned
Table4 the headings are not clear, the first column represent the variables or factors? The * means what? LV is not present you have to show its coefficient or intercept. So the second column is the estimate coefficient. Finally you have to mention the magnitudes and confidence intervals of the fixed effects; the magnitude of the among group variation for each random effect. Or report among group variation as random effect variances or proportions of variance. In the legend mention that these coefficient estimates come from LM o LMM analysis

Annotated reviews are not available for download in order to protect the identity of reviewers who chose to remain anonymous.
External reviews were received for this submission. These reviews were used by the Editor when they made their decision, and can be downloaded below.

---

## Round 0.2 · Minor Revisions

Dear authors,

Please revise the manuscript carefully including ALL the suggestions of the reviewers. If these are all included we could consider the manuscript for publication.

Reviewer 1 ·

Basic reporting

The manuscript has improved considerably, the authors have followed the comments and observations. But still there is some information to add, I didnt find the information of the soil properties per site, I think it is important to added,due to the fact that the authors make correlations with that information ant the earthworms abundance. Also I encourage the authors to add the information of Ngrass per site, it will be interesting if there were significant differences of Ngrass, among the sites. And also non in the text and non in the figures, tables, can I see the signficant differences indicated (a,b,c), I only saw the p value in the correlation table (Table 4).

Experimental design

The experimental design is better explained.

Validity of the findings

The results are interesting and well represented, but there is not information of the soil properties data per site, and the Ngrass data per site, only about the correlation of the earthworms community abundance and those variables, in the text it is not mention if there are signficant differences among the sites with those variables. In the previous version there was a table with soil properties per site, but that is not anymore in the newest version, so I suggests to add it. And Eventhough Anovas were performed, on results it is not indicate the signficance of the differences found among the sites in relation with the earthworms community abundances, and soil properties, Ngrass, climate parameters.

Additional comments

The manuscript has improved, but it is important to informe in the abstract which are the soil variables that are correlated with the earthworms community, in results:what are the signficant differences observed among the sites (using p values). Include information of soil properties and Ngrass also in this new version. Introduction and Discussion has improved considerebly, but please see the attached document.

Annotated reviews are not available for download in order to protect the identity of reviewers who chose to remain anonymous.

Reviewer 2 ·

Basic reporting

All my comments are in the general comments to the authors and in the manuscript

Experimental design

All my comments are in the general comments to the authors and in the manuscript

Validity of the findings

All my comments are in the general comments to the authors and in the manuscript

Additional comments

The manuscript has improved a lot. Many of the comment had been answered, although there is some points that still need to be clarified or discussed. The main ones are below
My previous comment on the presence of Pontoscolex in the Azores is due because the soil temperature is high because of volcanic activity coming from underground, although the air or environmental temperature is temperate. It should had been interesting to have data on soil temperature from the 5 sites. It is known that the soil is a buffer for the temperature changes. Maybe the soil form NA has a more buffer capacity to keep the soil warmer and that is why some Pontoscolex were found there.
My previous comment: “Something that you have not discussed and you should is: why in LV there is only in one monolith P. corethrurus and it is the site where the native species are, and they do not coexist with P.corethrurus. What this site has that P. corethrurus is almost not present although the climatic conditions are the best for it? Has not been clearly discussed”. Besides a clarification has to be done about the average abundance of P, corethrurus in LV. Part of the explanation is the aggregate distribution of P. corethrurus although it is very wired that it was found only in one point of the five from LV. Part of P. corethrurus great adaptability is its strong interactions with microorganisms for the digestion of soil organic matter (Lavelle et al 1995, Trigo et al Pedobiol 1999, Patron et al.)
The data of OM from NA and AC have to be checked, as I said in the manuscript these data are too high.
Your comment on lines279 -284 require more previous information of how the grasses are distributed in each sites and points. It is surprising that almost all the grasses are the same along the gradient. It is amazing that the only native grass is all along the gradient. The earthworm community is more sensitive to the altitudinal gradient than the grasses! The comment about the nitrogen and the soil deterioration do not reflect what is in table 2. For NA and AC the nitrogen content is high and the organic matter content is extremely high. The C/N ratio that I had calculated for these soils are as high as if they were pure litter or compost...
Minor comments or corrections are in the manuscript.

Annotated reviews are not available for download in order to protect the identity of reviewers who chose to remain anonymous.
External reviews were received for this submission. These reviews were used by the Editor when they made their decision, and can be downloaded below.

---

## Round 0.3 · accepted · Accept

The manuscript can be published in the present form

External reviews were received for this submission. These reviews were used by the Editor when they made their decision, and can be downloaded below.